# Epigallocatechin Gallate Modulates Muscle Homeostasis in Type 2 Diabetes and Obesity by Targeting Energetic and Redox Pathways: A Narrative Review

**DOI:** 10.3390/ijms20030532

**Published:** 2019-01-27

**Authors:** Ester Casanova, Josepa Salvadó, Anna Crescenti, Albert Gibert-Ramos

**Affiliations:** 1Nutrigenomics Research Group, Department of Biochemistry and Biotechnology, Universitat Rovira i Virgili (URV), Campus Sescelades, 43007 Tarragona, Spain; mariajosepa.salvado@urv.cat; 2Technological Unit of Nutrition and Health, EURECAT-Technology Centre of Catalonia, Avinguda Universitat 1, 43204 Reus, Spain; anna.crescenti@eurecat.org

**Keywords:** epigallocatechin gallate, obesity, muscle, oxidative stress, cell signaling

## Abstract

Obesity is associated with the hypertrophy and hyperplasia of adipose tissue, affecting the healthy secretion profile of pro- and anti-inflammatory adipokines. Increased influx of fatty acids and inflammatory adipokines from adipose tissue can induce muscle oxidative stress and inflammation and negatively regulate myocyte metabolism. Muscle has emerged as an important mediator of homeostatic control through the consumption of energy substrates, as well as governing systemic signaling networks. In muscle, obesity is related to decreased glucose uptake, deregulation of lipid metabolism, and mitochondrial dysfunction. This review focuses on the effect of epigallocatechin-gallate (EGCG) on oxidative stress and inflammation, linked to the metabolic dysfunction of skeletal muscle in obesity and their underlying mechanisms. EGCG works by increasing the expression of antioxidant enzymes, by reversing the increase of reactive oxygen species (ROS) production in skeletal muscle and regulating mitochondria-involved autophagy. Moreover, EGCG increases muscle lipid oxidation and stimulates glucose uptake in insulin-resistant skeletal muscle. EGCG acts by modulating cell signaling including the NF-κB, AMP-activated protein kinase (AMPK), and mitogen-activated protein kinase (MAPK) signaling pathways, and through epigenetic mechanisms such as DNA methylation and histone acetylation.

## 1. Structure, Bioavailability and Metabolism of Epigallocatechin Gallate

Epigallocatechin gallate (EGCG) is a major constituent of green tea (7380 mg per 100 g of dried leaves). Smaller amounts of EGCG are found in apple skin, plums, onions, hazelnuts, pecans and carob powder (at 109 mg per 100 g) [1]. Green tea, from the *Camellia sinensis* L., contains very high levels of flavan-3-ols monomers, also known as catechins, and its main components are (−)-epicatechin (EC), (−)-epigallocatechin (EGC), (−)-epicatechin gallate (ECG), and (−)-epigallocatechin gallate (EGCG). Green tea leaves are steamed to reduce oxidation, however, during the production of black tea, the levels of flavan-3-ols drop. The reason behind is that tea leaves are processed in a specific way that includes fermentation, which converts these flavan-3-ols in theaflavins and thearubigins [2].

### 1.1. Molecular Structure

The structure of EGCG consists of four rings resulting from the esterification of EGC with gallic acid: the A and C rings constitute the benzopyran ring with a pyrogallol moiety at position 2, the B ring, with a gallate moiety at position 3, and the D ring (Figure 1). The presence of this ester carbon makes EGCG highly susceptible to nucleophilic attack [3]. The B and D rings of EGCG have vicinal 3′, 4′, 5′ and 3″, 4″ and 5″-trihydroxy groups respectively, which give EGCG its anti-oxidative potential [3]. These ortho-dihydroxy pairings found in rings B and D account for EGCG’s potent divalent metal chelating capacity [4]. EGCG has been found to be a more efficient radical scavenger than its structural analogues EGC, EC, and ECG, which all have fewer hydroxyl groups [4]. The EGCG molecule is less stable in neutral and alkaline media because the hydroxyl groups on the phenyl ring are attacked by the basic medium, leading to the formation of a more active phenoxide anion. This instability results in low bioavailability [3].

### 1.2. Bioavailability and Metabolism

Many studies have shown a low systemic bioavailability of EGCG when it is taken orally. In an acute feeding study, healthy human subjects consumed 500 mL of green tea containing 648 µmol of flavan-3-ols, after which plasma and urine were collected over a 24-h period and analyzed by HPLC-MS [5]. The plasma contained a total of 10 metabolites in the form of *O*-methylated, sulfated, and glucuronide conjugates of EC and EGC, along with the native green tea flavan-3-ols EGCG and ECG [5]. The peak plasma concentration (Cmax) of unmetabolized EGCG was 55 nM and the time to reach Cmax (Tmax) was 1.6 h. Both the Tmax and the observed transformations are indicative of absorption in the small intestine [6]. The appearance of unmetabolized EGCG and ECG in plasma is unusual in dietary flavonoids and might be a consequence of the galloyl moiety inhibiting phase II metabolism [7,8].

While catechins are usually glucuronidated or sulfated in human plasma, EGCG can be found in free form and in high proportions (77–90%) [9,10,11,12]. Nevertheless, the fact remains that EGCG is poorly absorbed when administered orally due to its high solubility, resulting in low membrane permeability in human studies [4]. Moreover, EGCG is stable during gastric digestion at a low pH, but very unstable under duodenal conditions, in a more alkaline medium, which leads to low bioavailability.

The plasma levels of EGCG after intragastric administration of decaffeinated green tea to rats was found to be 0.1% [13]. Ullmann and colleagues reported that after the administration of 1600 mg EGCG to healthy volunteers, the Cmax was 3392 ng/mL [12]. Other human studies on the plasma kinetics of EGCG and its conjugated metabolites indicated that the total mean of EGCG area under the plasma concentration time curve between 0-h to infinity (AUC (0–N)) ranged from 442 to 10,368 ng·h/mL, and the mean terminal elimination half-life (t1/2z) was from 1.9 to 4.6 h when purified and isolated EGCG was supplemented to healthy individuals [12]. In addition, another study examined the plasma kinetics of purified EGCG after administrations of 800 mg once per day and 400 mg twice per day for 4 weeks [9]. A peak in the serum levels of EGCG was observed after it had been administered at 400 and 800 mg [9]. An unexpected high amount of bound EGCG was found when the affinity of EGCG for human serum albumin was tested in physiological conditions. These results imply that almost all EGCG is transported in the blood bound to albumin, and explains the wide tissue distribution and chemical stability of EGCG in vivo [14]. EGCG kinetics have also been studied in rat and mouse models. Male Sprague Dawley rats that were given 0.6% green tea in their drinking water for 14 days had an EGCG concentration in the large intestine of 487.8 ± 121.5 ng/g, while its concentration in the bladder was approximately of 201.4 ± 154 ng/g EGCG [15]. A few studies have shown that EGCG does indeed cross the blood-brain barrier [16,17]. In particular, male and female mice administered orally with 200 µL of 0.05% EGCG solution containing 3.7 MBq [^3^H]EGCG displayed 0.32% and 0.33%, respectively, of total radioactivity in the brain 24 h after uptake, which was comparable with most other organs [17]. Kohri et al. showed that after an oral administration of radioactive EGCG, its concentration in blood starts to increase after 8 h, shows a peak at 24 h, and then starts to decrease. In their study, they also show that major urinary excretion happened during the increase and peak periods, and that at 72 h its excretion was 32.1% of the oral dose, while its excretion in the feces during the following 72 h was 35.2% of the dose [18]. Kohri and colleagues also studied the metabolic fate of EGCG in rats supplemented with antibiotics and [4-^3^H]EGCG, and found that the excretion levels were lower than in normal rats, and so, concluded that the radioactivity observed in the blood and urine came from EGCG degradation by the microbiota. Additionally, the authors report that a particular metabolite in the normal rats was purified and identified as 5-(5′-hydroxyphenyl)-γ-valerolactone 3′-*O*-β-glucuronide (M-2), while in feces, EGC (40.8% of the fecal radioactivity) and 5-(3′,5′-dihydroxyphenyl)-γ-valerolactone (M-1, 16.8%) were detected [18]. They propose that M-1 was absorbed in the body after EGCG was degraded by intestinal bacteria, yielding M-1 with EGC as an intermediate. Additionally, M-2 could have been formed from M-1 in the intestinal mucosa and/or liver, then entered the systemic circulation, and finally excreted in the urine [18].

Urine excreted 0 to 24 h after green tea consumption, contained a profile of flavan-3-ol conjugates similar to the plasma; however, ECG and EGCG were undetectable [18]. These outcomes indicate that the intact flavan-3-ols, ECG and EGCG do not undergo extensive metabolic modifications. Several researchers have observed that it is not possible to detect EGCG in urine, despite its presence in plasma [9,19,20], which is difficult to explain. It could be that the kidneys are unable to absorb EGCG from the plasma; however, if this is the case, there must be other mechanisms that produce its rapid decline after Cmax is reached [21]. Auger et al. [22] supplemented patients with an ileostomy with pure EGCG from green tea and analyzed the ileal fluid and urine over a period of 24 h. They did not find EGC or its metabolites in urine, thus establishing that degalloylation did not occur endogenously [21]. It has been hypothesized in animal studies that EGCG may be cleared from the plasma in the liver and returned to the small intestine through the bile [23,24]. Although this enterohepatic recirculation is not yet proven in humans, it might be possible that this EGCG from the bile is degallated by the gut microbiota, and then, if no more degradation happens, it is excreted in urine as EC and EGC metabolites [6].

Most of the consumed flavan-3-ols after green tea ingestion reach the large intestine where they are modified by the microbiota. These successive modifications result in their transformation to C-6–C-5 phenylvalerolactones and phenylvaleric acids, followed by their conversion into C-6–C-1 phenolic and aromatic acids, which are absorbed by the colon, enter the bloodstream, and are excreted in urine. The total amount that finally exits the body through urine is approximately a third of the ingested flavan-3-ols [21]. Moreover, it is believed that these transformations that occur in the colon might have important bioactive effects, and that these effects might vary between individuals because of differences in the microbiota composition. Even so, further studies are needed to make such assumptions [25].

It has been found, comparing two acute green tea feeding studies, one with volunteers subjected to an ileostomy [26] and the other with healthy individuals [5], that there were almost no differences in the plasmatic flavan-3-ol pharmacokinetics. These results indicated that flavan-3-ol monomers are principally absorbed in the upper part of the gastrointestinal tract [6]. Even so, it was also reported in the study with human subjects with an ileostomy that the ileal fluid contained 70% of the initially supplemented green tea flavan-3-ols [26]. Therefore, it can be stated that, in healthy subjects, the major part of favan-3-ols consumed will pass from the small to large intestines [6].

## 2. Muscle in Obesity: The Problem of Inflammation

Obesity is a term defined by the World Health Organization (WHO) that involves an abnormal or excessive amount of body fat accumulation that presents a risk to health [27]. Nowadays, obesity is becoming a global public health issue because affected individuals are at a major risk of developing a big range of comorbidities such as cardiovascular disease, type 2 diabetes (T2D) and respiratory disorders, among others [28,29].

Recent insights in obesity indicate that the adipose tissue exerts an inflammatory influence on the body. The metabolic effects of inflammation include insulin insensitivity, hyperlipidemia, muscle protein loss and oxidant stress [30]. Activation of the immune system increases the production of oxidant molecules. Moreover, oxidants could take part in the inflammatory response activating the nuclear factor kappa-B (NF-κB) which is linked to many of the genes related to the inflammatory response [31]. White adipose tissue derivatives, either fatty acids (FA) or inflammatory cytokines, in obesity have many adverse and synergistic effects on the skeletal metabolism [32]. High levels of interleukin-6 (IL-6), an inflammatory cytokine, are associated with insulin resistance (IR) and T2D [33,34,35] likely due to greater white adipose tissue IL-6 secretion [36,37].

In obesity and T2D subjects, the IL-6 signaling pathway in muscle cells has an abnormal function because there is a reduction in the expression of IL-6 receptor and also an abnormal STAT3/suppressor of cytokine signaling 3 (SOCS3) in adipose tissue [35,38]. SOCS3 is involved in the inhibition of leptin and insulin signaling, and has been found to be elevated in the skeletal muscle of mice fed with a high fat diet [39] and in insulin resistant states [40]. Overexpression of SOCS3 impairs leptin-stimulated AMPK activation, reduces tyrosine phosphorylation of IRS-1, PI-3-kinase activity, and AKT phosphorylation [39]. Thus, the reduction of SOCS3 expression in most cases could improve leptin and insulin resistance [41].

In addition, the lipid excess observed in obesity damages the muscle, causing cellular dysfunctions. Indeed, lipid excess in skeletal muscle activates endoplasmic reticulum (ER) stress and, consequently, the accumulation of unfolded or misfolded proteins in the ER lumen. Furthermore, it has been shown that the muscles of obese insulin-resistant individuals contain almost 30% less mitochondria than normal individuals [42], which suggests that muscles have less capacity to oxidize fatty acids, and thus contribute to insulin resistance.

Currently, there are many studies focusing on the treatment of the systemic inflammation induced by obesity and T2D, and particularly, through the use of natural bioactive compounds [43,44], which are usually harmless and more secure than synthetic drugs [45]. Moreover, targeting obesity-induced inflammation would be useful for the treatment of obesity and related diseases, since this state is linked to many illnesses or problematics, such as an overall activation of the immune system [46], cancer [47] or metabolic diseases [48].

## 3. EGCG on Energy Metabolism: Animal Models and Human Studies

In recent years, the interest in the health benefits of dietary components for preventing obesity and T2D has increased [49]. Specifically, polyphenol compounds such as a green tea extract rich in EGCG exert anti-obesogenic properties [50,51]. Preclinical studies on animals and some studies on humans have confirmed the beneficial effects of EGCG [29] on obesity-related parameters including decreased body weight [52,53,54], decreased adipose mass [52], reduction of food intake [55], decreased total lipids, cholesterol and triglyceride in the liver and plasma, and an improvement in glucose homeostasis [52,53,56]. Various mechanisms have been proposed to produce these responses, including suppression of dietary fat absorption [52,57,58], enhancement of fat oxidation in adipose tissue and skeletal muscle [53,59], increase of glucose utilization [52,60,61], and decrease of de novo lipogenesis [59,62,63].

Moreover, data also provide evidence that green tea extracts have beneficial effects improving body composition and weight [53,64,65], reducing body fat [53,60,61], improving glucose and lipid metabolism [53,61,65], and protecting against ER stress, oxidative stress and protein degradation induced by high fat diet in skeletal muscle [66].

Treatment of C57b1/6J mice with 0.32% dietary EGCG for 16 weeks has been shown to reduce body weight gain and markers of T2D [52] induced by a high fat diet. In obese KK-ay mice, EGCG reduces ROS content, decreases glucose levels and increases glucose tolerance in animals [67]. Some of these effects are mediated by epigenetic mechanisms because, one of the mechanisms of EGCG appears to be the direct inhibition of DNA methyltransferases (DNMT) [68,69].

Laboratory studies with animal models have generally demonstrated that green tea and EGCG play a role in the prevention of obesity and have beneficial effects on glucose homeostasis, oxidative stress and lipid metabolism [49]. However, the effects on humans have been less studied. Epidemiological studies have suggested some of the possible effects of EGCG on humans, but there are few controlled intervention studies and many of these have different methodological designs. Therefore, more studies are needed to elucidate the action mechanisms [70]. A few studies have been carried out on humans with the aim of determining whether EGCG and/or green tea extract mediates the effects of lipid metabolism or energy expenditure in skeletal muscle. A study [71] researched the effect of 3-day supplementation of 282 mg/day EGCG on overweight subjects and found a non-significant effect on skeletal muscle lipolysis. It also found decreased lactate concentration, which suggests a shift towards a more oxidative muscle phenotype and also indicates that a longer period is necessary for the prevention of obesity [71,72]. Additionally, a systematic review on the effects on metabolic parameters such as respiratory quotient and energy expenditure concluded that EGCG could have a positive effect on both parameters, however, the authors conclude that further and larger prospective trials are needed [73].

The findings on humans suggest that EGCG alone also has a potential effect on fat oxidation. A randomized, double-blind, placebo-controlled, crossover pilot study showed that the administration of 300 mg EGCG/day for two days decreased the respiratory quotient during the first postprandial phase, suggesting an increase in fat oxidation and a potential anti-obesity effect [74]. Another study conducted with human subjects reported a decrease in body weight and body fat as well as an increase in fat oxidation and thermogenesis; these findings were confirmed in cell culture systems and animal models of obesity [63]. Other studies and designed experiments with green tea consumption combined with resistance training aimed to determine whether the observed beneficial effects were increased with exercise. Results demonstrated that there was a decrease in body fat, waist circumference and triglyceride levels, and an increase in body mass and muscle strength [75]. In contrast, in a randomized controlled trial study, overweight or obese male subjects randomly took 400 mg capsules of EGCG twice a day over 8 weeks. The results conclude that EGCG had no effect on insulin sensitivity, insulin secretion or glucose tolerance measured in blood extractions [50]. Therefore, well-designed and controlled clinical studies are necessary to validate the results of these human studies. Moreover, it must be stated that the concentrations used in many animal studies or the extrapolation of the doses from cell culture experiment into those equivalents in humans, greatly surpass what would be considered a physiological dose in humans. This might explain the differences in observed results between the different models and should be taken into account in future studies.

## 4. Metabolic Effects of EGCG in the Muscle and Muscle Cell Lines

### 4.1. Mitochondria and Oxidative Stress

The main role of mitochondria is energy production. For this reason, an impairment of their function could be implicated in insulin resistance and obesity [76]. The oxygen available for the respiratory chain activity may undergo incomplete reduction giving rise to ROS that is scavenged by the antioxidant defenses of the organelle. Likewise, there are several other ROS producers that have been identified in muscle cells that are activated by different stimuli including nicotinamide adenine dinucleotide phosphate (NADPH), oxidases (NOXs), phospholipase A2 (PLA2), xanthine oxidase (XO) and lipoxygenases [77]. Moreover, ROS can also be produced from non-muscle sources, such as immune cells, in response to muscle injures from exercise [78]. Furthermore, it has also been proposed that ROS activates ERK and/or JNK and induces autophagy in skeletal muscle [79]. The overproduction of ROS could lead to oxidative stress. To protect against this oxidative stress, enzymatic and non-enzymatic antioxidant systems could regulate ROS. The enzymatic systems, which include superoxide dismutase, glutathione peroxidase and catalase, have the capacity to convert ROS into less active molecules and prevent the transformation of these less active species into a more deleterious form. However, non-enzymatic antioxidants, such as EGCG, can protect against this undesirable effect and prevent pathological states that involve oxidative cell damage [80]. Indeed, the intake of EGCG by Wistar rats also reveals a decrease in plasma markers of oxidative stress, and an increase in antioxidant enzymes. Thus modulating and protecting from the effects of oxidant species by increasing antioxidant enzymatic systems and also due to their antioxidant capacity [81].

In skeletal muscle, the oral gavage of 100 mg EGCG/kg/day for 3 months in diabetic rats significantly reduced the expression levels of Beclin1 and dynamin-related protein 1 (DRP1) [79]. This implies that EGCG regulated mitochondrial-involved autophagy and ameliorated excessive muscle autophagy through down regulation of the ROS/ERK/JNK-p53 pathway [79]. Muscle is a major site of ATP production and energy consumption where the uptake and oxidation of glucose and fatty acids are key molecular events [64]. Moreover, EGCG has also been reported to inhibit mitochondrial oxidative phosphorylation to decrease ATP levels [82]. All these actions would result in an increase in the ADP/ATP ratio to activate AMPK, although these possibilities are still being researched.

### 4.2. Endoplasmic Reticulum

It has been shown that a high fat and sucrose diet, apart from dysregulation of lipid homeostasis, can also activate endoplasmic reticulum (ER) in skeletal muscle [83,84]. ER is an organelle that regulates a variety of post-transcriptional protein modifications, so its disruption leads to the accumulation of unfolded or misfolded proteins in the ER that affects a variety of cellular signaling processes, including energy production, inflammation and apoptosis. Several studies suggest that altered redox homeostasis in the ER causes ER stress and induces ROS production in ER and mitochondria [85,86]. If the ER stress is prolonged and cannot be restored, the functional homeostasis of the ER is impaired, which can induce an inflammatory state, protein degradation and cell death [85,87]. In mice fed with a high fat diet and receiving green tea extract rich in EGCG for 20 weeks, Rodriguez [66] found that there was a protective effect in muscle against oxidative stress and ER stress due to the repression of the increase in Binding immunoglobulin protein (BiP), Activating transcription factor 4 (ATF4), X-box binding protein 1s (XBP1s) and X-box binding protein 1u (XBP1u) in skeletal muscle protecting the cell from apoptosis and cell death.

### 4.3. Lipids

Other studies have also shown that EGCG may modulate muscle lipid metabolism by increasing lipid use, suggesting that thermogenesis and fat oxidation are increased [60] and, thus, that EGCG has lipid lowering properties [88]. Human and animal studies have revealed that excessive lipid accumulation in addition to obesity could also lead to insulin resistance or heart failure. Therefore, this lipid accumulation deteriorates cell signaling pathways and causes cellular dysfunctions [66,89]. It has been shown that chronic feeding of green tea extract could be implicated in fatty acid oxidation; elevating gene expression factors involved in lipid transport and oxidation such as fatty acid translocase (FAT)/CD36, medium-chain acyl-CoA dehydrogenase (MCAD) and uncoupling protein 3 (UCP3) [49,60]. Another study demonstrated that the treatment of 1% dietary EGCG for 4 weeks in mice reduced high fat diet-induced increase in body weight and body fat mass as well as increased mRNA expression of UCP2 and UCP3 in liver and skeletal muscle. These two genes are related to fatty acid oxidation, explaining the effects of EGCG on body weight gain. EGCG in this experiment also downregulates genes related to fatty acid synthesis and storage in the liver and white adipose tissue [54,90]. Moreover, β-oxidation in the gastrocnemius muscle of lean mice, fed with green tea, was elevated and exercise capacity was increased, as well as the expression of FAT/CD36 mRNA in skeletal muscle [60]. Friedrich [91] studied the acute effect of EGCG on oxidation and fat depots, and showed that in male C57BL/6 mice fed with different high fat diets, and an EGCG supplement, there was an increase of postprandial dietary fat oxidation and a decrease of dietary fat incorporation in skeletal muscle. These results were correlated with the downregulation of lipogenesis genes (ACC, FAS, SCD1) and lipid synthesis in liver and skeletal muscle in the postprandial state after 2 and 4 days of treatment.

### 4.4. Glucose

The role of EGCG in the modulation of glucose uptake and disposition has also been studied. Skeletal muscle is a key regulator of glucose homeostasis and contributes to postprandial blood glucose levels [92]. Skeletal muscle accounts for about 75% of insulin-stimulated whole-body glucose uptake [93]. To stimulate glucose uptake in the muscle cells, insulin promotes the translocation of GLUT4 from intracellular storage vesicles to the plasma membrane [94]. Many studies have focused on the regulation of the expression of genes involved in glucose uptake and insulin signaling to determine the mechanism behind the antidiabetic activities of green tea [57]. An experiment with mice fed with a high fat diet and treated with 0.32% EGCG for 16 weeks found a reduction in body weight and insulin resistance and an increase in the mRNA levels of the nuclear respiratory factor (Nrf1), medium chain acyl coA decarboxylase (Mcad), UCP3 and peroxisome proliferator α (PPAR-α) related to lipid metabolism [49]. Serisier et al. [95] carried out an experiment with dogs and showed that green tea can affect insulin sensitivity. Obese, insulin-resistant dogs were treated orally with green tea extract (80 mg/kg bw per day) over 12 weeks. The results showed that insulin resistance was decreased by 20% and in skeletal muscle the expression of PPARα and LPL mRNA increased; however, GLUT4 mRNA did not increase. In contrast, an experiment on Wistar rats with 0.1–0.2% dietary green tea treatment over 6 weeks concomitant with a high fructose diet showed that mRNA GLUT4 and IRS1 levels increased in muscle [96]. Similar results were found in an experiment carried out with male C57BL/6J mice fed with a high-fat diet and green and black tea supplements, where body weight gain and fat deposition in white adipose tissue were suppressed due to the stimulation of glucose uptake and the upregulation of the GLUT4 expression in the plasma membrane of muscle cells [97]. Experiments on Wistar rats revealed that EGCG attenuated free fatty-induced insulin resistance by activating the AMPK pathway [81]. Enhanced GLUT4 translocation and activation of AMPK in skeletal muscle have also been observed in lean mice treated with oolong, black or Pu-erh tea (2% extract as drinking fluid) for 7 days [98]. AMPK is a phosphorylating enzyme that regulates metabolic homeostasis of fatty acid and glucose metabolism in skeletal muscle and the inhibition of adipocyte lipolysis and the modulation of insulin secretion by pancreatic β-cells [99]. For this reason, there is interest in developing AMPK activators as a potential therapy for diabetes and obesity [94,100,101]. This AMPK activation would decrease gluconeogenesis and fatty acid synthesis while increasing catabolism, leading to a reduction in body weight [64]. Therefore, the effects of green tea rich in EGCG on glucose metabolism associated with T2D seems to be mediated in various ways, including glucose production, increased insulin secretion and insulin sensitivity, and increased uptake of glucose into skeletal muscle [102]. Various authors have discussed these mechanisms and their beneficial effects [103,104]. Indeed, experiments carried out on rodents have detected several other mechanisms attributed to the decrease on body weight. EGCG may modulate energy expenditure by inhibiting catechol-o-methyltransferase (COMT), which would lead to increased fat oxidation [105,106]. Another mechanism is the involvement of EGCG in sirtuins activation, especially Sirtuin 1 (SIRT1), which deacetylates histones and non-histone proteins including transcription factors [107]. The activation of SIRT1 in various types of polyphenols, such as EGCG, is beneficial for reducing energy absorption and increasing fat oxidation in diet-induced obesity in mice [54]. Moreover, SIRT1 is associated with the transcriptional co-factor peroxisome proliferator-activated receptor-y-coactivator 1 α (PGC1 α) activation, thereby also improving the mitochondrial function and protecting against metabolic diseases [107,108].

In experiments with isolated myocytes, it was found that EGCG stimulates GLUT4 translocation and results in an increased glucose uptake [56,61,97]. A study in male Wistar rats with an intake of EGCG during 3 weeks, also ameliorated glucose homeostasis [92]. EGCG acts as a potent antioxidant and protects against oxidant agents such as docosahexaenoic acid (DHA) in in vitro experiments by decreasing ROS levels and restring changes in mitochondrial morphology induced by DHA [109]. Moreover, other experiments with mouse C2C12 muscle cells treated with EGCG have shown a reduction in ROS levels [110,111], and a beneficial effect on fatty acid-induced peripheral insulin resistance [112]. There are many mechanisms involved in such response including: inhibition of PKC activation enhanced by the AMPK cascade, IRS1 serine phosphorylation and other kinases such as ERK1/2 and p38 MAPK, essential for maximal stimulation of glucose uptake in response to insulin [113,114]; and finally, the suppression of lipid accumulation via the AMPK/ACC signaling pathway [112,115]. In mouse C2C12 myotubes, the inhibition of AMPK following glutamate dehydrogenase (GDH) activation was reversed by EGCG. GDH senses mitochondrial energy supply, and its stimulation in primary human myotubes caused lowering of insulin-induced 2-deoxy-glucose uptake, which was partially counteracted by EGCG. Thus, mitochondrial energy provision, through anaplerotic input via GDH, influences the activity of the cytosolic energy sensor AMPK [116]. In addition, an in vitro experiment with L6 cells with insulin resistance and treated with 20 µM EGCG, demonstrated an improvement in glucose uptake by GLUT 4 translocation to the plasma membrane [94,117], which depended on the key regulator AMPK [118] and PI3-K/Akt activation [56,117,119]. Other studies on EGCG reported that it stimulates the MAPK pathway and the Nrf2 transcription factor leading to the transcription activation of the ARE-mediated Phase II genes induction [120].

Some of EGCG effects are mediated by epigenetic mechanisms because EGCG inhibits directly DNMT by interacting with the catalytic site of the DNMT molecule [121]. Studies with structural analogues of EGCG suggest the importance of D and B ring structures in the inhibitory activity. Molecular modelling studies also support the direct inhibitory effect of EGCG on DNMT; EGCG forms hydrogen bonds with Pro(1223), Glu(1265), Cys(1225), Ser(1229) and Arg(1309) in the catalytic pocket of DNMT [68]. EGCG also acts by inhibiting histone acetyl transferase activity and NF-κB activation [122].

## 5. Future Approaches

As exposed in this review, there are numerous studies focusing on EGCG and its effects on the many metabolic disturbances that affect the skeletal muscle in obesity and T2D, however, more research must be done to identify the specific metabolic pathways implicated on its response or the appropriate treatment for human patients. Moreover, the low systemic bioavailability of EGCG raises doubts about the feasibility of this molecule for the treatment or prevention of particular diseases.

In order to overcome the problem of EGCG instability and low bioavailability, several studies have tried the use of nanocarriers, such as liposomes and gold nanoparticles, which have shown to greatly increase the effects of EGCG in particular diseases [123]. On the other hand, scientists are studying the possibility of combining different bioactive ingredients with known beneficial activities on the causes of obesity or the inflammatory state. In this sense, there are studies focusing on the combination of these ingredients in order to obtain additive or synergic effects that could increase the overall effect of the individual ingredients [124,125]. Specifically for EGCG, it has been shown that combining EGCG with other bioactive ingredients with similar activities, and its supplementation to obese individuals, improved many of the obesity related parameters [126,127].

## 6. Conclusions

In conclusion, EGCG may be useful to treat obesity by re-sensitizing insulin-resistant muscle, increasing muscle lipid oxidation and stimulating the glucose uptake of muscle cells. Likewise, EGCG may revert the increase of ROS production, ER stress and protein degradation in skeletal muscle in obesity by regulating mitochondrial-involved autophagy and ameliorating excessive muscle autophagy (Figure 2). Despite being a potent antioxidant, EGCG does not seem to act in vivo as conventional hydrogen-donating antioxidants due to its low bioavailability. The circulating EGCG concentration is in the nanoM interval, similar to hormones. Thus, rather than acting as a chemical antioxidant, in vivo EGCG generates signals for inducing protective enzymes and may exert modulatory actions in cells acting as signaling molecules and/or regulating gene expression. In this sense, EGCG, among other mechanisms, up-regulates genes encoding for phase II enzymes with antioxidant response elements (AREs) in the promoter region; has insulin-mimetic actions via the PI3K/Akt signaling pathway; activates AMPK, an energy-responsive and redox sensing enzyme; and inhibits DNA methyltransferases and histone acetyl transferases, thus mediating epigenetic mechanisms.

It would be interesting to apply these results to counteract obesity, where EGCG could be used as a complement to sport programmes and good eating habits to obtain better responses against obesity.

## Figures and Tables

**Figure 1 ijms-20-00532-f001:**
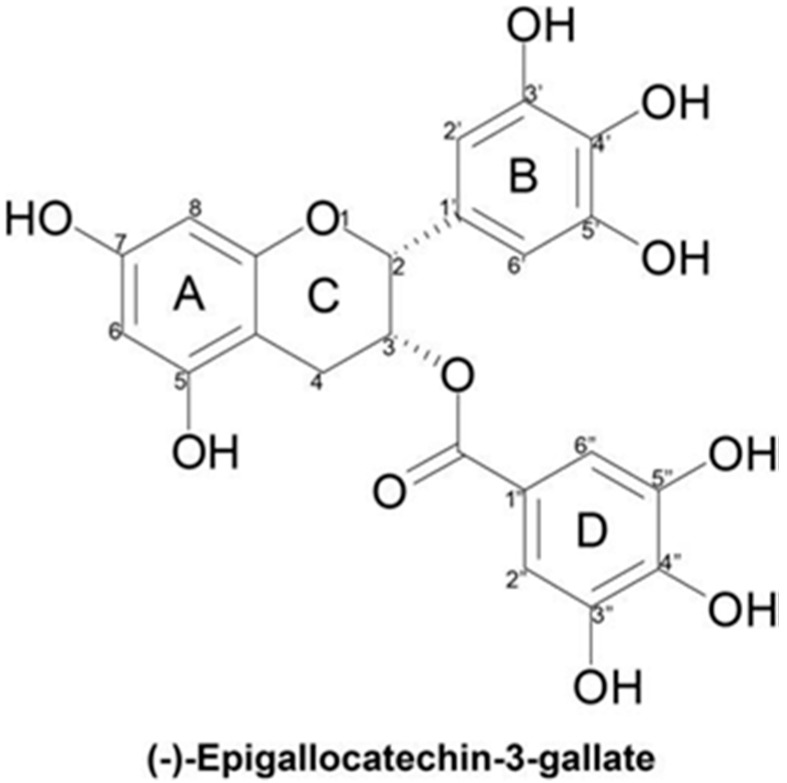
Chemical structure of (−)-Epigallocatechin-3-gallate. The four rings resulting from the esterification of epigallocatechin (EGC) with gallic acid are indicated by letters (A, B, C and D) and each carbon from each ring is indicated by numbers (1–8).

**Figure 2 ijms-20-00532-f002:**
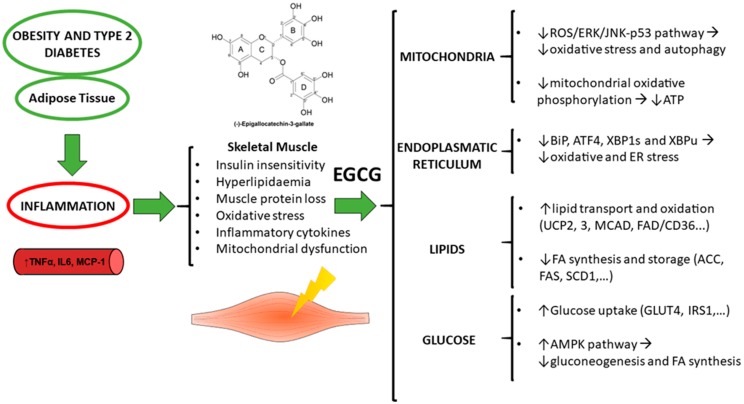
Summary of the effects of (−)-Epigallocatechin-3-gallate on the modulation of muscle homeostasis in obesity and type 2 diabetes. Arrows indicate an increase (↑) or decrease (↓) of that particular molecule or the pathway’s activity.

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
