# Peer review of "Epigallocatechin Gallate Modulates Muscle Homeostasis in Type 2 Diabetes and Obesity by Targeting Energetic and Redox Pathways: A Narrative Review"

_ijms, 2019, doi:10.3390/ijms20030532_

Round 1

Reviewer 1 Report

The work is now acceptable for publication in this reviewer's opinion.

Reviewer 2 Report

Authors are fully respected and made corrections suggested by the reviewers. I have no further comments. I thank that the manuscript is acceptable for publication.

Reviewer 3 Report

The authors well addressed to the comments. This reviewer has no more comment.

This manuscript is a resubmission of an earlier submission. The following is a list of the peer review reports and author responses from that submission.

Round 1

Reviewer 1 Report

The work is well written, with only minor language editing needed. It is a "narrative" review, and I would add the term "narrative" in the title, as this shouldn't be mistaken for a systematic work. 

I would perhaps also add a short discussion about the concentrations used in cell models and about the doses used in animal studies, that most of the times vastly exceed what could be possibly be considered physiological in humans.

Finally, I would definitely add a summary figure, explaining in a nice graphical way what are the mechanisms putatively involved in EGCG effects. In the end, reviews are often great material for lectures and talks, and I do not really appreciate them if they do not a dd a clear and cutting figure. 

Reviewer 2 Report

This paper is well-written with a logical sequence but it lacks some of the chapter-specific EGCG mechanisms that would contribute to the overall effectiveness of EGCG in the modulation of homeostasis of type 2 diabetes and obesity.

I suggest to update the work as follows:

1.      The efficiency and mechanisms of EGCG would be more apparent through the Figure that shows molecular mechanisms of the anti-obesity effects of EGCG

2.      Add a subtitle: The effect of EGCG on reduced intake of food through the mechanism of regulation of the neuropeptide

3.      Add  s subtitle:  Interaction among EGCG, obesity and gut microbiota. This chapter should includ an increase in energy harvest, modulation of free fatty acids especially butyrate of bile acids, lipopolysaccharides, gamma-aminobutyric acid (GABA), an impact on toll-like receptors, metabolic endotoxinemia and metabolic infection. How the metabolism of EGCG and its metabolites affects on the growth of Lactobacillus, Enterococcus spp. and Bifidobacterium spp,  since their amount in obesity significantly decreases. In addition, impact of sex hormones on the development of gut microbiota.

4.      The effect of EGCG in resolving inflammation through activity of M1 and M2 macrophage

5.      the role of inflammatory pathways in metabolic and hemodynamic dysfunction, cardiac and vascular injury

6.      Some data on EGCG's ability to regenerate after muscle injury, especially myeloid leukocyte migration, regulation of tumor necrosis factor production, CD4-positive, alpha-beta T cell differentiation, ECM organization, and toll-like receptor (TLR) signaling. Moreover, changes in complement activity. It would be nice to see the role of T cells, especially the regulatory T cells that can prime other immune cells, and also exhibit various functions related to the regulation of body weight, insulin-resistance and glucose tolerance.

7.      The relationship between obesity-induced inflammation and clinical possibilities of targeting inflammation, as well as better use of EGCG. Proposed possible research to increase EGCG efficacy and effectiveness in modulating homeostasis type 2 diabetes and obesity.

8.      Minor correction: Plese add number of references (Auger et al.). Plese separate SI units from number (20µM)

Reviewer 3 Report

This manuscript is basically well-written, but it need some revision. Please address following points.

1. The authors mixed up the previous results from EGCG (compound level) and tea or tea extract (food and food material) experiments. Although EGCG is a major polyphenol in tea, the authors should clearly distinguish both of them to avoid misunderstanding for the readers.

2. As the authors wrote in the manuscript, bioavailability of EGCG is low and the levels of its aglycon form in the body is limited. However, the authors wrote that ‘only polyphenol known to be present in plasma in a free form’ in the Abstract section. Epicatechin shows the most highly bioavailability among chatechins as you know. Thus, this sentence should be deleted.